# The Role of Sleep Patterns from Childhood to Adolescence in Vigilant Attention

**DOI:** 10.3390/ijerph192114432

**Published:** 2022-11-04

**Authors:** Efrat Barel, Orna Tzischinsky

**Affiliations:** Department of Behavioral Sciences and the Center for Psychobiological Research, The Max Stern Academic College of Emek Yezreel, Emek Yezreel 1934000, Israel

**Keywords:** sleep, children, adolescents, vigilant attention, PVT

## Abstract

Only a few studies addressed age-related changes from childhood to adolescence in sleep patterns, sleepiness, and attention. Vigilant attention plays a key role in cognitive performance. While its nature and course have been investigated broadly among adults, only limited research has been conducted on its development between childhood and adolescence. The main aim of the current study was to replicate previous findings about the effects of sleep loss on age-related changes in vigilance attention performance and sleepiness in a natural setting. A total of 104 children and adolescents (46 children aged 6–9 and 58 adolescents aged 13–19) wore an actigraph for a continuous five to seven nights, including weekdays and weekends. Subjective sleepiness (Karolinska Sleepiness Scale) and a Psychomotor Vigilance Test (PVT-B) were measured on two school days and one non-school day. Findings showed that PVT-B performance differed by age group, with adolescents outperforming children in PVT-B measures in spite of their elevated subjective sleepiness. Adolescents demonstrated less sleep time and increased sleepiness. Although PVT-B performance was better among adolescents, a within-subject analysis revealed that adolescents performed better on PVT measures on weekends than on weekdays. The results are discussed in relation to the synaptic elimination model.

## 1. Introduction

Sleep time, sleep duration, and circadian rhythms change dramatically across human development, especially from childhood to adolescence. However, sleep patterns are marked by individual differences. Therefore, it is important to consider individual differences in sleep as a factor underlying individual differences in cognitive and brain development [1,2]. While a recently published review [1] dealt mainly with memory and less with attention, the current study links children’s and adolescents’ development and changes in sleep with the issue of vigilant attention.

Vigilant attention is a core cognitive function that underlies other cognitive domains, such as learning, memory, and executive functioning [3,4]. It refers to the attention control processes needed to preserve attention and task engagement over time [5] and plays a critical role in daily functions. Deficits in vigilant attention are present across a wide range of psychiatric and neurological illnesses [6]. While the nature and course of vigilant attention have been primarily investigated among adults, vigilant attention development between childhood and adolescence has not yet been fully characterized.

The existing studies documenting age-related changes in vigilant attention have yielded inconsistent findings, with some showing poorer performance in childhood and aging populations than in adulthood [7], some showing no difference in performance between younger and older adults [8], and others showing improved performance in older adults [9]. A few studies suggested that vigilant attention performance follows a U-shaped, curvilinear trend. Fortenbaugh et al. [10] investigated the vigilant attention performance of 10,430 participants and found it to rapidly improve between the ages of 10 and 16, moderately improve throughout adulthood, and decline in later adulthood. Similarly, McAvinue et al. [7] described the poor performance in childhood and adolescence, a plateau across adulthood, and a decline in later adulthood.

Several studies focused on age-dependent performance in vigilant attention among young populations, documenting the rapid improvement in vigilant attention from childhood to adolescence. For example, Lewis et al. [11] showed that children aged 10–11 performed higher than children aged 6–8 in all sustained attention measures (reaction time, commission, and omission errors). Functional neuroimaging studies of pediatric populations demonstrated that increased activation in mainly right-lateralized brain areas during sustained attention tasks was positively correlated with increasing age [12].

Levels of neurobehavioral performance, including vigilant attention, are determined by both endogenous and exogenous processes [13]. Among the endogenous factors, the circadian clock and sleep homeostasis interact in humans to ensure high and stable performance during awakening [14,15]. In young adults, sleep deprivation results in reaction time and lapses increase [16,17]. With the transition from childhood to adolescence, there is a decrease in sleep time, increased sleepiness, and a desire for more sleep among adolescents [18,19,20]. In addition, adolescents suffering from sleep deprivation showed impaired daytime functioning, worse school performance and achievements [21,22,23,24], and attention problems, particularly increases in lapses of attention in vigilant attention tasks [25,26]. Among the few studies addressing sleep and attention in childhood, Vriend et al. [27] investigated sleep in relation to daytime functioning among children aged 8–12 and found that sleep efficiency was correlated with poorer scores on divided attention. Spruyt et al. [28] examined the role of sleep in the attentional performance of children and adolescents aged 6–18 in a natural setting. They found that across development, various sleep parameters, such as sleep duration, sleep regulation, and sleep midpoint, become increasingly important for attentional performance.

Only a few studies addressed age-related changes in attention and sleepiness. In a longitudinal study investigating the influence of sleep duration on daytime sleepiness and vigilant attention among participants aged 10–16, Campbell et al. [29] documented a paradoxical pattern: while daytime sleepiness increased with age, vigilant attention improved with age. The authors used synaptic elimination [30] to interpret the results in relation to the model of adolescent brain development. Synaptic pruning was found to influence two measures differentially. In adolescence, the intensity of brain activity creates a high need for sleep-dependent recuperation, and the restriction of sleep creates a greater increase in sleepiness. In contrast, synaptic elimination reduces the redundancy of neural pathways, a brain reorganization that may explain the improvement in vigilance performance.

Although some changes in sleep timing and duration occur during puberty, the processes controlling sleep regulation change drastically in adolescence. The 2-process model [31] suggests sleep regulation by homeostatic sleep and circadian factors. Homeostatic sleep pressure increases throughout the day (with increased time awake), while the circadian rhythm works to keep humans active and alert until the night, helping them maintain optimal sleep/wake alignment with the environmental patterns of dark/light [32]. Apart from their influence on sleep timing, these two processes may also modulate neurobehavioral functions during awakening, which can affect cognitive performance, including attention [33].

Even though studies addressed the developmental trajectory of vigilant attention performances, only a few controlled for sleep factors in this relation. Given that impairment in reaction time task performance is induced by sleep loss among adults [14] and appears to change across the life span, we employed a cross-sectional design to examine age-related changes in vigilant attention across a wide age range of children and adolescents (ages 6–19). Our home-based study evaluated the impact of sleep on vigilant attention in a natural setting through a sleep/wake pattern. We compared several weekday nights with weekend nights. While most studies have used self-report measures for both sleep quality and cognitive performance, this study assesses sleep quality via objective measures (using actigraphy) and vigilant attention (using a PVT test on an iPad). By conducting the study in a natural setting, we were able to follow the developmental shift in sleep patterns from childhood to adolescence and examine age-related changes in vigilant attention in connection with sleep. We investigated whether our results obtained from cross-sectional data replicate longitudinal data obtained from restricted time in bed schedules regarding PVT performance and sleepiness among children and adolescents. In particular, based on the model of adolescent brain development and synaptic elimination [29], we hypothesized that adolescents would outperform children in vigilant attention, even following nights of sleep loss, as measured by attention lapses and reaction time.

## 2. Materials and Methods

### 2.1. Participants

Study participants included 104 children and adolescents: 46 children (out of 50, 4 stopped participation due to the outbreak of the COVID-19 pandemic; 23 females; mean age 7.5 years, range 6–9 years) and 58 adolescents (out of 60, 2 stopped participation; 31 females; mean age 16.3 years, range 13–19 years) from normative elementary, middle, and high schools in northern Israel. Partial results from the older subsample have been previously reported [34] (the association between sleep and neurobehavioral performance). In the current study, we added a group of children to the previous study and employed the same study design in order to test age-related changes in vigilant attention. Data from each group were collected separately. The age range in each group was chosen according to the relationship between sleep and puberty [35]. For the children group, we looked for children before puberty, according to parents’ reports. For the adolescents, we looked for adolescents who had passed the stage of puberty development according to their parents’ reports. On average, girls begin puberty at 10–11 years old and boys at 11–12 years old [36].

### 2.2. Materials

Cognitive performance measures. Participants completed a visual Psychomotor Vigilance Test (PVT-B) three times a day: in the morning after wake-up, in the afternoon between 2 and 4 p.m. (after school), and before bedtime. This task is sensitive to sleep loss [37]. Despite the PVT being sensitive to the circadian phase [38], in our study, no main effect was found for time or interactions including time. Therefore, scores were averaged across the day. This simple three-minute-long measure of sustained attention reaction time (RT) to repetitive stimuli has become recognized as a highly sensitive and effective tool for measuring the degradation of sustained attention performance under partial sleep deprivation or changes in circadian phases [39]. The PVT-B (Joggle Research Program, Seattle, WA, USA) was performed here on an iPad. Participants were instructed to maintain vigilant attention on a target box and to respond as quickly as possible to the appearance of a stimulus while avoiding premature responses. The outcome measures of the present study were mean RT and lapses. Lapses are defined as ≥500 ms latency between stimulus presentation and participant response. Participants were requested to press on the screen to stop the counter, responding as quickly as possible but avoiding pressing on the screen when the counter was not displayed (i.e., false starts). The validated three-minute version of the PVT-B has inter-stimulus intervals ranging from 1 to 4; the longer inter-stimulus interval (i.e., up to 10 s) is associated with the 10-minute PVT [37]. It is important to note that although a short test may be less sensitive to detect fatigue, the PVT-B version was judged to show acceptable sensitivity and specificity for the assessment of fatigue [37]. Nevertheless, the short (3 min) test is more appropriate for young children and adolescents than the long test (10 min).

### 2.3. Sleep Measures

Objective sleep patterns were measured using an actigraph (Mini-Act, AMA-32, AMI), a small motion sensor that continuously records body motility data for long periods. The actigraph was worn on the non-dominant wrist for a seven-night period and accumulated movement. Sleep/wake measures were estimated using a validated algorithm [40,41]. The validation results indicated that for healthy subjects, the agreement rates between actigraph-based and PSG-based minute-by-minute sleep–wake scoring was above 90% [42]. The actigraph measured: sleep onset time, wake time, sleep efficiency ([total sleep time/total time in bed] × 100), sleep duration, wake after sleep onset (WASO: the number of minutes during the sleep period scored as awake), and sleep latency. Sleep loss was defined as sleep duration below the recommended range [43]. For children (6–13 years), 9–11 h of sleep per day is recommended, whereas, for adolescents (14–17 years), 8–10 h of sleep per day is recommended. In our sample, children slept the appropriate sleep durations both on weekends and weekdays. Adolescents, on the other hand, slept less than recommended on weekdays (see Table 1).

Finally, the Karolinska Sleepiness Scale (KSS) [44] was also used here. The KSS is a scale consisting of 9 statements relating to sleepiness. Scores range from 1 (extremely alert) to 9 (extremely sleepy), with higher scores indicating greater subjective sleepiness. The KSS is used very often among various age groups, including children [45], and has been shown to be correlated with objective measures of sleepiness, e.g., [46]. Participants filled in the KSS questionnaire three times a day: in the morning after wake-up, in the afternoon after school between 2 and 5 p.m., and before bedtime. The scores were averaged per day.

### 2.4. Procedure

The College Institutional Review Board approved the complete study protocol (EMEK-YVC:2018-48). All parents of participants signed an informed consent form before the experiment. Actigraph, iPads, and questionnaires, along with a detailed explanation, were given to the participants at home by research assistance. The data collection lasted about six months per group.

### 2.5. Statistical Methods

Continuous data were represented by mean and standard deviation. Approximate normality was assessed via evaluation of skewness and kurtosis: the variable was considered approximately normal if skewness and kurtosis were between −1.0 and +1.0. As sleep latency was not normally distributed, we analyzed the log.

Weekday wake time was approximately normally distributed after removing one subject with a wake time of 10.17. Weekday WASO was approximately normally distributed after removing one subject with WASO = 140.6.

Group differences of the categorical background variables were assessed by χ^2^ tests. A mixed models analysis of variance was performed on the sleep data for the model, with a group (children/adolescents) and day type (weekdays/weekends) as the main effects and the interaction between them using unstructured repeated measures covariance. This was repeated without the outliers to verify the results. Post hoc testing of the interactions was evaluated via a Student’s t-test using Bonferroni corrections. In order to compute an effect size, Cohen’s f2 was computed by taking the difference between two models: one with the variable of interest and one without (small effect = 0.02, medium = 0.15, large = 0.35). We then calculated R2 for the two models and computed Cohen’s f2. Cohen’s d was computed for the group’s main effect. A similar analysis was performed for PVT lapses.

As PVT reaction time was not normally distributed, we used a gamma distribution for model reaction time. Repeated measures correlations were performed using Rmcorr macro in R (version 4.1) to assess the association between sleep and performance. Repeated measures correlations determine commonality within individual associations for paired measures assessed multiple times per subject. Bonferroni corrections for multiple comparisons were calculated. All other analyses were performed using SPSS (version 24). Significance was set at *p* < 0.05.

## 3. Results

Table 1 presents the objective sleep data by group and day type. A mixed models analysis of variance revealed group differences for all sleep measures except for sleep latency and significant day-type differences for sleep onset time, wake time, and sleep efficiency. Moreover, there was a statistically significant group-by-day-type interaction for wake time and sleep duration.

Across all day types, children went to bed on average 2.2 h earlier (22:10 vs. 24:27), woke up 41 min earlier (7:29 vs. 8:10), and slept 96 min more (560.0 vs. 463.7 min) than adolescents. In addition, children spent significantly fewer minutes awake after sleep onset than adolescents (28.7 vs. 46.3 min). Across all participants, on the weekends, sleep onset time was 1:21 later, and wake time was 1:40 later than on weekdays (23:59 vs. 22:38, 8:40 vs. 6:59, respectively). It should be noted that there was no statistically significant difference in sleep duration between weekends and weekdays (521 vs. 503 min). Sleep efficiency was lower on the weekends than on weekdays (87.2% vs. 89.5%). Analysis of the group-by-day-type interaction revealed that there were group differences in wake time and sleep duration on both weekdays (F(1, 95) = 4.13, *p* < 0.05, F(1, 95) = 143.85, *p* < 0.001; respectively) and weekends (F(1, 82) = 9.03, *p* < 0.004, F(1, 82) = 13.53, *p* < 0.001; respectively). Within each group, there was a statistically significant difference in wake time between weekdays and weekends (children: F(1, 37) = 45.51, *p* < 0.001; adolescents: F(1, 45) = 69.89, *p* < 0.001). However, there was a statistically significant difference in sleep duration between weekdays and weekends in the adolescent group (F(1, 48) = 8.53, *p* < 0.005) but not in the children group (F(1, 37)= 0.44, *p* > 0.51). When the analyses of wake time and WASO were repeated without the outliers, the results remained the same. However, when sleep efficiency below 50% was removed, there was no difference in sleep efficiency between weekdays (89.5%) and weekends (88.2%).

For subjective sleepiness (KSS), a significant main effect for the group was found (F(1, 98) = 20.81, *p* < 0.001) with adolescents being sleepier than children (5.0 vs. 4.0). There was also a group-by-day-type interaction for sleepiness (F(1, 102) = 4.58, *p* < 0.04). Further analyses revealed a statistically significant difference in sleepiness between weekdays and weekends in the adolescent group (5.3 vs. 4.7, t(54) = 3.78, *p* < 0.001) but not in the children group (4.01 vs. 3.97, t(44) = 0.88, *p* > 0.05). Adolescents were sleepier on weekdays than on weekends.

### Performance

As expected, there was a statistically significant difference in PVT performance measures between the two age groups (see Table 2). Children had slower reaction times (M = 395.79, SD = 16.23) and greater lapses (M = 11.72, SD = 0.99) than their adolescent counterparts (M = 268.98, SD = 12.97; M = 4.86, SD = 0.78, respectively). There was also a significant day-type main effect for reaction time with faster reaction times on the weekend (M = 327.89, SD = 13.14) than on weekdays (M = 336.87, SD = 9.44). There was a group-by-day-type interaction for lapses. Post hoc testing revealed that there were statistically significantly fewer lapses among adolescents on the weekend than on weekdays (3.9 vs. 5.8; F(1, 43) = 24.10, *p* < 0.001) but no difference among children (12.03 vs. 11.41; F(1, 40) = 0.29, *p* > 0.59).

Table 3 presents the repeated measures correlation. Overall, participants’ wake time was negatively correlated with PVT lapses: the earlier the participants woke up, the greater the number of lapses. However, when the outlier was removed, participants’ wake time did not correlate with PVT lapses.

Among the children, sleep was not correlated with performance. Among the adolescents, mean PVT reaction time was negatively correlated with sleep onset time and wake time: the earlier the participants went to sleep, and the earlier they woke up, the higher the reaction time. In addition, PVT lapses among adolescents were negatively correlated with sleep onset and wake times, as well as sleep duration. Moreover, KSS was positively correlated with PVT reaction time and lapses. However, after Bonferroni corrections for multiple comparisons, only the correlations between PVT lapses and sleep duration and between KSS and mean PVT reaction time and lapses among adolescents remained significant.

## 4. Discussion

In the current study, PVT performance differed by age group both in reaction time and lapse measurements, with adolescents outperforming children in both PVT measures. These findings support previous studies showing rapid improvement in vigilant attention throughout childhood and adolescence [10]. Among the studies addressing the developmental trajectory of vigilant attention, only a few objectively measured sleep quantity and quality. In our study, objective sleep duration was correlated with PVT performance but only in adolescents. Furthermore, although children and adolescents differed in objective and subjective sleep measures, PVT performance was better among adolescents. Adolescents demonstrated increased sleepiness and greater gaps in objective sleep measures between weekdays and weekends. However, their performance on weekdays with fewer sleep hours (partial sleep deprivation defined as sleep duration below the recommended range; [43]) was still better than children’s performance. These paradoxical results are in accordance with Campbell et al.’s [29] findings about increases in both daytime subjective sleepiness and daytime vigilant attention among adolescents. Campbell et al. [29] interpreted their results in relation to the model of adolescent brain development by synaptic elimination [30]. Recent studies have also reported associations between sleep microstructure and brain maturation. In a cross-sectional sample (ages 2–26), Kurth et al. [47] found that topography of maximal SWA predicted cognitive performance and gray matter maturation. Specifically, in adolescence, as synaptic elimination proceeds, the intensity of brain activity declines, leading to increased sleepiness. At the same time, synaptic elimination reduces redundant neuronal pathways, which contributes to improvements in processing speed. A recent meta-analysis examined neural correlates to vigilant attention in children and adolescents [12]. The study reviewed functional neuroimaging studies in pediatric populations and showed that neurodevelopmental changes in brain maturation occur throughout childhood and adolescence. During adolescence, decreased segregation and increased integration contribute to the efficient integration of information between different functional domains of the brain and the greater capability to facilitate higher-order cognitive functions [48].

Although adolescents outperformed children in PVT measures even following sleep loss, a comparison of their performance on weekdays and weekends showed higher performance on weekends. Campbell et al. [29] suggested that the increase in synaptic pruning may decrease processing speed but may also increase vulnerability to sleep loss. It has been proposed that sleep loss can reduce redundant neuronal pathways below the level required for optimal cognitive performance [49]. These effects of sleep loss on PVT performance may be due to the multidimensional features of attention [37], including variability in the maintenance of an alert state [50], selective attention [42], orienting network, and executive network [51,52]. Sleep loss influences PVT performance, causing slower response time and increasing numbers of lapses can be understood through the state instability theory [50]. According to Doran, Van Dongen, and Dinges [53], the competing systems of sleep initiation (the involuntary drive to fall asleep) versus wake maintenance (the top-down drive to sustain alertness) lead to unstable sustained attention under conditions of sleep loss.

Despite the current study’s contributions, certain limitations should be noted. First, by being conducted in a natural setting, it is not, in essence, so experimental. Given the reduced internal validity of such a setting, findings should be interpreted with caution. Nevertheless, the setting enables an imitation of the “real world” effects of partial sleep deprivation, which thus strengthens external and ecological validity. Due to the need for changes in public opinion and policy about the impact of sleep loss on adolescents, we recommend future studies use a longitudinal home experimental setting with large sample sizes, and wide age ranges in order to uncover the developmental trajectory of PVT performance from childhood through adolescence and into early adulthood. In addition, while we are aware of the validity problem of the PVT-B, neither children nor adolescents in the real world are able to perform the PVT for a long time (10 min) unless they are in a laboratory.

A second limitation relates to sample characteristics. The children and adolescents participating in the current study were healthy and relatively good sleepers [43]. The extent to which our findings can be generalized to other populations is, therefore, unknown. Third, in a recent meta-analysis of reviews, Spruyt [54] suggested operationalizing the multilevel concept of sleep through various forms. Although the present study included multiple characteristics of sleep quality and quantity, including objective and subjective measures, there is still a need for a comprehensive operationalization of sleep. Future studies should address additional factors, such as chronotype, in order to deepen our understanding of age-related changes in attentional performance across childhood development. Finally, the present findings are restricted to the short-term consequences of sleep loss on PVT performance. Future studies should examine the long-term consequences of sleep loss on both brain structure and function among children and adolescents.

In sum, the current study investigated the effects of sleep loss on age-related changes in PVT performance from childhood to adolescence in a natural setting. Adolescents outperformed children in PVT measures in spite of their elevated subjective sleepiness. Examining the effect of sleep loss within each age group only revealed an effect for adolescents with improved performance due to increased sleep duration. These paradoxical results were also reported by Campbell et al. [29] and supported by neuroimaging studies showing age-dependent functional networks [12]. Furthermore, while brain maturation through synaptic elimination may account for the increased performance in vigilant attention, it may also reinforce the vulnerability to sleep loss [29].

## 5. Conclusions

Our results suggest that age is associated with PVT performance, showing increasing age to be associated with better performance on PVT measures. However, in a further analysis within each age group, we found that adolescents performed better on weekends in comparison to weekdays, stressing the need to implement the recommended sleep duration for optimal performance. In order to establish recommendations for sleep duration, future studies should examine neural correlates of vigilant attention across a wide age range and use various vigilant attention modalities.

## Figures and Tables

**Table 1 ijerph-19-14432-t001:** Sleep data by group and day type. Data are mean ± SD [range].

	Children (N = 46)		Adolescent (N = 58)		Group				Day Type				Group by Day			
	Weekday	Weekend	Weekday	Weekend	F	DF	*p*	Cohen’s d	F	DF	*p*	Cohen’s f2	F	DF	*p*	Cohen’s f2
Onset time (h)	21.45 ± 0.60 [20.27–22.80]	22.85 ± 1.34 [20.37–27.13]	23.80 ± 1.08 [22.10–26.80]	25.11 ± 1.78 [21.87–29.77]	100.87	1.94	0.001	1.361	79.62	1.88	0.001	0.159	0.24	1.88	0.63	0.002
Wake time (h)	6.84 ± 0.46 [5.87–8.10]	8.11 ± 1.16 [6.12–11.27]	7.13 ± 0.82 [5.82–10.17]	9.23 ± 1.76 [5.73–12.97]	12.214	1.94	0.001	0.706	116.22	1.88	0.001	0.085	6.71	1.88	0.01	0.008
Duration (min)	564.1 ± 43.7 [445.0–631.0]	556.3 ± 74.7 [356.0–747.0]	441.0 ± 54.8 [246.0–529.5]	486.5 ± 92.7 [229.0–684.0]	77.77	1.92	0.001	0.810	3.47	1.90	0.07	0.002	7.17	1.90	0.009	0.006
Sleep efficiency (%)	95.3 ± 3.2 [87.7–100.0]	94.3 ± 4.0 [82.5–99.6]	83.7 ± 5.0 [64.3–92.7]	80.7 ± 13,8 [21.0–98.0]	84.17	1.81	0.001	2.671	4.93	1.84	0.03	0.029	2.04	1.84	0.16	0.019
Sleep latency (min)	18.16 ± 12.39 [0.00–71.80]	14.71 ± 11.18 [0.00–56.00]	20.13 ± 15.47 [1.50–65.40]	34.19 ± 39.19 [0.00–159.0]	1.24	1.94	0.27	0.102	0.69	1.90	0.41	0.001	2.24	1.90	0.14	0.011
WASO (min)	26.76 ± 17.94 [1.6–65.6]	31.14 ± 22.49 [3.0–97.0]	46.89 ± 20.58 [14.2–140.6]	46.33 ± 28.55 [2.0–92.3]	20.01	1.96	0.001	1.035	0.24	1.88	0.62	0.000	0.83	1.88	0.36	0.004

Notes: F = F statistics from mixed models analysis of variance; DF = degrees of freedom; (numerator, denominator); Cohen’s d, Cohen’s f2 = model effect size.

**Table 2 ijerph-19-14432-t002:** Performance (PVT) data by group and day type. Data are mean ± SD [range].

	Children (N = 46)		Adolescent (N = 58)		Group				Day Type				Group by Day Type			
	Weekday	Weekend	Weekday	Weekend	F	DF	*p*	d	F	DF	*p*	f2	F	DF	*p*	f2
Mean RT (s)	460.99 ± 211.06 [250.57–1457.00]	440.14 ± 202.84 [276.03–1368.30]	278.03 ± 48.27 [250.57–1457.00]	266.16 ± 49.86 [199.03–419.20]	68.30	1.88	<0.001	1.5	5.99	1.62	0.02	0.004	1.40	1.56	0.24	0.002
Lapses (no.)	13.01 ± 6.15 [1.33–28.67]	13.35 ± 6.85 [3.50–30.05]	5.71 ± 4.53 [0.50–20.63]	4.11 ± 3.83 [0.00–15.50]	62.54	1.92	<0.001	1.6	3.73	1.81	0.06	0.025	8.25	1.82	0.005	0.051

Notes: F = F statistics from mixed models analysis of variance; DF = degrees of freedom; (numerator, denomonfirmedinator); Cohen’s d, Cohen’s f2 = model effect size.

**Table 3 ijerph-19-14432-t003:** Repeated measures correlation of sleep with PVT mean RT and lapses for all participants and by age group.

	Mean RT		Lapses	
	r (95% CI)	*p*	r (95% CI)	*p*
Onset	−0.02 (−0.12, 0.08)	0.674	−0.07 (−0.17, 0.03)	0.185
Children	0.02 (−0.13, 0.17)	*0.805*	*0.08 (0.07, 0.23)*	*0.302*
Adolescents	−0.22 (−0.35, −0.08)	*0.002*	*−0.26 (−0.39, −0.13)*	*0.004*
Wake time	−0.02 (0.12, 0.08)	0.698	−0.14 (−0.24, −0.04)	0.006
No outlier	−0.02 (−0.12,0.08)	0.702	0.13 (−0.03, 0.27)	0.149
Children	0.05 (−0.10, 0.20)	*0.914*	−0.22 (−0.43, 0.01	*0.062*
Adolescents	−0.26 (−0.38, −0.12)	*0.004*	−0.40 (−0.51, −0.28)	*0.004*
Sleep duration	−0.0011 (0.10, 0.10)	0.983	−0.11 (−0.21, −0.01)	0.027
Children	*0.04 (−0.12,0.18)*	*0.648*	*0.06 (−0.09, 0.21)*	*0.447*
Adolescents	*−0.13 (−0.26, 0.02)*	*0.082*	** *−0.30* ** *(−0.42, −0.17)*	*0.001*
Sleep efficiency	0.02 (−0.08, 0.12)	0.707	0.02 (−0.08, 0.12)	0.693
Children	*−0.00 (−0.15, 0.15)*	*0.987*	*−0.06 (−0.22 0.08)*	*0.362*
Adolescents	*0.12 (−0.03, 0.26)*	*0.120*	*0.10 (−0.05, 0.24)*	*0.190*
Sleep latency (Log)	−0.06 (−0.16, 0.04)	0.295	−0.04 (−0.14, 0.06)	0.425
Children	*−0.10 (−0.24, 0.05)*	0.199	*−0.07 (−0.25, 0.08)*	*0.370*
Adolescents	*0.01 (−0.13, 0.15)*	0.891	*−0.0 (−0.16, 0.12)*	*0.792*
WASO	0.03 (−0.07, 0.13)	0.801	0.01 (−0.09, 0.12)	0.587
No outlier	0.01 (−0.02, 0.12)	*0.808*	0.02 (−0.02, 0.13)	0.677
Children	*0.03 (−0.12, 0.18)*	0.669	*0.09 (−0.06, 0.24)*	*0.241*
Adolescents	*−0.02 (−0.16, 0.12)*	0.786	*−0.02 (−0.16, 0.12)*	*0.768*
KSS	0.03 (−0.07, 0.12)	0.603	0.07 (−0.02, 0.17)	0.138
Children	−0.03 (−0.17, 0.11)	*0.683*	−0.05 (−0.19, 0.10)	*0.513*
Adolescents	**0.31** (0.24, 0.48)	** *0.000* **	**0.280** (0.15, 0.40)	*0.000*

Note: Bold values indicate the only correlation that remained significant after performing Bonferroni corrections for multiple comparisons.

## Data Availability

The data that support the findings of this study are available from the corresponding author upon request.

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
