# Peer review of "The Role of Sleep Patterns from Childhood to Adolescence in Vigilant Attention"

_ijerph, 2022, doi:10.3390/ijerph192114432_

Round 1

Reviewer 1 Report

This paper entitled “The role of sleep patterns from childhood to adolescence in vigilant attention” described the relationship between sleep and attention in a developmental view. The authors compared a set of sleep parameters as well as attention (using PVT) in different groups (children vs adolescents) and days (weekdays vs weekends). The authors found that adolescents performed much better than children in PVT test, despite that the adolescents slept less. Finally, a synaptic elimination model may be served as a neural mechanism for this phenomenon.

The experimental design of this paper is clear, and the topic of this study is crucial for the field. I have the following comments to improve the current version:

1.     There are a few parameters used in this study that need further explanation in the Methods chapter: for example, what is “WASO” and how to calculate? for? How to calculate “sleep efficiency”?

2.     Line 163: change “group” to “group(children/adolescents)”

3.     Line175: change”p<.05” to “p<0.05”

4.     Line 186: Does “1:21” mean “1 hour 20 mins?”

5.     Line 187:it’s a bit confuse, what are compared here?

6.     Table2 : what are the units for “MeanRT and Lapses”

7.     Table3: “Children Adolescents” need to be separated into two rows in the Sleep efficieny, also the fonts of these numbers are different. Moreover, is it necessary to specify p value into 1-4? Would be clearer to highlight the data that have positive/negative relationships.

Author Response

First Reviewer:

  • We provided calculation explanations in the Methods section for WASO and sleep efficiency (p.4).
  • We changed "group" to "group (children/adolescents)" (p.4).
  • We changed ".05" to "0.05" (p.5).
  • 1:21 means 1 hour and 21 minutes (p.5)
  • We rephrased the sentences in order to clarify the comparison between weekdays and weekends (p.5).
  • We added the units for mean RT and lapses in Table 2 (p.6).
  • We separated "children and adolescents" into two rows (p.7).
  • We added p-values to the table 3 to avoid confusion (p.7).
  • Following corrections of the p-values used for the calculations in Table 3, we highlighted the correlations which remained significant (p.7).

Reviewer 2 Report

This work aimed to examine the role of sleep patterns on vigilant attention from childhood to adolescence by using Karolinska Sleepiness Scale (KSS) and a Psychomotor Vigilance Test (PVT-B) based on the few studies of aged-related changes in attention and sleepiness. The topic is interesting. Some comments for the authors to improve the quality of the manuscript are below.

1. There is a problem with the format of the abstract. What problem was this study asking based on what research context? What was trying to be solved? What conclusions were ultimately drawn? Could the author give some explanations?

2. In line 78, what was 2-process model? What was its relevance to this study? Could the author give some explanations?

3. In line 107, what variables were involved in the previous experiment? Could this be briefly described again in tabular form?

4. In the part of Materials and Methods, how were the study participants in each age group categorized? Which age group in the sample was the most vigilant? What was the duration of the experiment? Why was the 10-12 years old study sample not included in the study? Could the author give some explanation?

5. In line 143, why was the Karolinska Sleep Scale preferred to be used over other scales in this research? State the reasons for its use in the article.

6. In Table 1, what did each of these letters mean? It is better to label them at the bottom of the table.

7. In line 235, formatting issues: distancing the content of the markup from the content of the text.

8. There is no conclusion at the end of the article, does it need to be added?

Author Response

Second Reviewer:

  • We corrected the phrasing regarding the objective of the present study in the Abstract (p.1).
  • We addressed the relevance of the two-process model to the present study (p.2).
  • We specified the variables investigated in a previous study (p.3).
  • In the Methods section we elaborated on how participants in each group were categorized (p.3).
  • In the Methods section we addressed the duration of data collection (p.4).
  • We provided an explanation as well as a reference regarding the exclusion of participants aged 10–12 years old (p.3).
  • We provided an explanation for the use of the KSS (p.4).
  • We labeled the letters appearing in Table 1 at the bottom of the table (p.5).
  • We addressed the formatting issue (p.7).
  • We added a conclusion at the end of the article (p.9).

Round 2

Reviewer 2 Report

Most of the questions have been revised, only need further attention in terms of language cohesion and graphical presentation.